# Pharmacokinetic/Pharmacodynamic Target Attainment Based on Measured versus Predicted Unbound Ceftriaxone Concentrations in Critically Ill Patients with Pneumonia: An Observational Cohort Study

**DOI:** 10.3390/antibiotics10050557

**Published:** 2021-05-11

**Authors:** Matthias Gijsen, Erwin Dreesen, Ruth Van Daele, Pieter Annaert, Yves Debaveye, Joost Wauters, Isabel Spriet

**Affiliations:** 1Clinical Pharmacology and Pharmacotherapy, Department of Pharmaceutical and Pharmacological Sciences, KU Leuven, 3000 Leuven, Belgium; erwin.dreesen@kuleuven.be (E.D.); ruth.vandaele@uzleuven.be (R.V.D.); isabel.spriet@uzleuven.be (I.S.); 2Pharmacy Department, UZ Leuven, 3000 Leuven, Belgium; 3Uppsala Pharmacometrics Research Group, Department of Pharmacy, Uppsala University, 751 05 Uppsala, Sweden; 4Drug Delivery and Disposition, Department of Pharmaceutical and Pharmacological Sciences, KU Leuven, 3000 Leuven, Belgium; pieter.annaert@kuleuven.be; 5BioNotus, Galileilaan 15, 2845 Niel, Belgium; 6Laboratory for Intensive Care Medicine, Department of Cellular and Molecular Medicine, KU Leuven, 3000 Leuven, Belgium; yves.debaveye@uzleuven.be; 7Laboratory for Clinical Infectious and Inflammatory Diseases, Department of Microbiology, Immunology and Transplantation, KU Leuven, 3000 Leuven, Belgium; joost.wauters@uzleuven.be

**Keywords:** ceftriaxone, critically ill, pharmacokinetic, community-acquired pneumonia, severe CAP, target attainment, protein binding, unbound concentration

## Abstract

The impact of ceftriaxone pharmacokinetic alterations on protein binding and PK/PD target attainment still remains unclear. We evaluated pharmacokinetic/pharmacodynamic (PK/PD) target attainment of unbound ceftriaxone in critically ill patients with severe community-acquired pneumonia (CAP). Besides, we evaluated the accuracy of predicted vs. measured unbound ceftriaxone concentrations, and its impact on PK/PD target attainment. A prospective observational cohort study was carried out in adult patients admitted to the intensive care unit with severe CAP. Ceftriaxone 2 g q24h intermittent infusion was administered to all patients. Successful PK/PD target attainment was defined as unbound trough concentrations above 1 or 4 mg/L throughout the whole dosing interval. Acceptable overall PK/PD target attainment was defined as successful target attainment in ≥90% of all dosing intervals. Measured unbound ceftriaxone concentrations (CEFu) were compared to unbound concentrations predicted from various protein binding models. Thirty-one patients were included. The 1 mg/L and 4 mg/L targets were reached in 26/32 (81%) and 15/32 (47%) trough samples, respectively. Increased renal function was associated with the failure to attain both PK/PD targets. Unbound ceftriaxone concentrations predicted by the protein binding model developed in the present study showed acceptable bias and precision and had no major impact on PK/PD target attainment. We showed suboptimal (i.e., <90%) unbound ceftriaxone PK/PD target attainment when using a standard 2 g q24h dosing regimen in critically ill patients with severe CAP. Renal function was the major driver for the failure to attain the predefined targets, in accordance with results found in general and septic ICU patients. Interestingly, CEFu was reliably predicted from CEFt without major impact on clinical decisions regarding PK/PD target attainment. This suggests that, when carefully selecting a protein binding model, CEFu does not need to be measured. As a result, the turn-around time and cost for ceftriaxone quantification can be substantially reduced.

## 1. Introduction

Community-acquired pneumonia (CAP) is a major reason for hospital admission [1,2,3]. Five to forty percent of the patients hospitalized with a diagnosis of CAP require admission to the intensive care unit (ICU), which is then commonly defined as severe CAP [4]. Ceftriaxone is a first-line antimicrobial often prescribed for the empirical treatment of severe CAP. Despite improved survival with ceftriaxone therapy, mortality rates are still very high, ranging from 25% to more than 50% [3,5].

For β-lactam antibiotics, pharmacokinetic/pharmacodynamic (PK/PD) target attainment, defined as time above the minimal inhibitory concentration (MIC), has been linked to positive clinical outcomes in critically ill patients [6]. However, as a result of large PK variability in critically ill patients, standard dosing regimens for several β-lactams have been shown to lead to suboptimal PK/PD target attainment [6,7]. Therefore, dose optimization strategies, such as therapeutic drug monitoring (TDM), are increasingly being performed in the ICU. Next to meropenem and piperacillin, ceftriaxone is one of the most frequently monitored β-lactams in the ICU [8]. Studies have shown suboptimal PK/PD target attainment in septic [9] or general ICU [10,11,12] patients treated with ceftriaxone. Nevertheless, routine TDM for ceftriaxone has still not found its way into clinical practice [7,13]. This may be due to the fact that ceftriaxone PK alterations and the impact on protein binding and PK/PD target attainment still remain unclear in specific populations.

The evaluation of the unbound concentration, which is the pharmacologically active fraction, is essential when assessing PK/PD target attainment [14]. For many β-lactams, the unbound concentration can be accurately predicted based on published protein binding percentages. Compared to most β-lactams, ceftriaxone shows remarkable features. Ceftriaxone shows extensive albumin binding (83–96% in healthy volunteers) and a long half-life, explaining the standard once daily dosing regimen [15]. In the ICU setting, where hypoalbuminemia is present in up to 50% of all patients, the unbound ceftriaxone fraction is highly variable [15,16,17]. This is probably due to its nonlinear concentration-dependent protein binding [10,18]. Measuring the unbound fraction using equilibrium dialysis, which is the reference method, is time-consuming and costly [19,20]. Therefore, there have been several attempts to estimate unbound ceftriaxone concentrations (CEFu), according to a fixed percentage of protein binding or to a predictive protein binding model [21]. Still, the impact of using protein binding models for ceftriaxone on PK/PD target attainment has not been investigated. As a result, most recent publications measure CEFu, for which ultrafiltration is mostly used, as this is less time-consuming and costly than equilibrium dialysis [11,18].

In this study, we aimed to evaluate if PK/PD target attainment of CEFu measured by equilibrium dialysis was acceptable in patients admitted to the ICU with severe CAP. Besides, we built a predictive model for ceftriaxone protein binding to investigate if CEFu could be reliably predicted from total concentrations and if the prediction of CEFu leads to different conclusions regarding PK/PD target attainment. 

## 2. Results

### 2.1. Patients and Ceftriaxone Concentrations

Thirty-one patients were included, contributing 72 samples over 36 dosing intervals (Figure 1). Most patients were sampled during a dosing interval either on an early (*n* = 20) or a late sampling day (*n* = 6), while five patients were sampled on both sampling days. During four of these 36 dosing intervals, the second sample was taken at midterm due to practical reasons, hence no trough sample was available then. As a result, 32 trough and 40 non-trough concentrations were collected during this study. In each sample, both CEFt and CEFu were measured. Overall, 50 samples were taken on the early day and 22 samples on the late day. Patient characteristics are shown in Table 1.

Overall, the median measured total ceftriaxone concentration (CEFt) and CEFu were 78.6 mg/L (range 5.1–327 mg/L) and 14.1 mg/L (range 0.5–85 mg/L), respectively. No concentrations were below the lower limit of quantification. The median measured unbound fraction was 17% (range 5.3–50%). At trough, the median measured CEFt and CEFu were 30.5 mg/L (range 5.1–123 mg/L) and 3.8 mg/L (range 0.5–15.9 mg/L), respectively. The median measured unbound fraction at trough was 13% (range 5.3–22%).

### 2.2. Protein Binding Model 

A model with nonlinear protein binding, including albumin as a significant covariate and BPV on Bmax best described CEFt concentrations and protein binding:(1)$$\mathrm{CEFt}=\mathrm{CEFu}+\mathrm{CEFu}\times \frac{\left(\mathrm{Bmax}\times {\left(\frac{\mathrm{ALB}}{0.44}\right)}^{h}\right)}{\left(B50+\mathrm{CEFu}\right)}$$
where CEFt stands for total ceftriaxone concentration (mmol/L), CEFu stands for unbound ceftriaxone concentration (mmol/L); Bmax stands for maximum binding capacity (mmol/L); ALB stands for albumin concentration (mmol/L), *h* stands for hill coefficient, and B50 stands for dissociation constant (mmol/L).

The VPC graph (Appendix A) shows that the measured CEFu fall within the range of predicted CEFu values.

This model was then transformed to predict CEFu from CEFt, resulting in the following equation:(2)$$\mathrm{CEFu}=-0.64\times \left({\left(\frac{\mathrm{ALB}}{0.44}\right)}^{0.26}\right)\times \mathrm{CEFt}+{\mathrm{CEFt}}^{1.09}$$

Model parameters for this formula, as well as for the other predictive models used in this study are reported in Appendix A).

### 2.3. Agreement between Measured CEFu and Predicted CEFu

The CEFu, unbound fraction and bias according to fixed average protein binding, saturable concentration-dependent protein binding and protein binding predicted from three different mixed-effects models (present study, Bos and Leegwater) are shown in Table 2. The relative bias for the various models is illustrated in Figure 2. Except for CEFu values predicted using the model developed in the present study and the Leegwater formula, all other predicted CEFu values presented statistically significant bias (*p* < 0.0001). When considering only trough levels, only CEFu values predicted using the model developed in the present study and the Bos formula did not show statistically significant bias. However, for all protein binding models, except for the model developed in this study, precision was low, as demonstrated by the high relative RMSE values.

### 2.4. Ceftriaxone PK/PD Target Attainment

The overall PK/PD target attainment was <90%, as both 100% *f*T_>MIC_ and 100% *f*T_>4xMIC_ were attained in 26 (81%) and 15 (47%) of 32 trough samples, respectively. From the covariates described in Table 1, only age and CrCl_CG_ were significantly correlated with both targets (*f*T_100%>MIC_ and *f*T_100%>4xMIC_). Younger patients with higher CrCl_CG_ showed higher rates of failure to attain PK/PD targets. Age was significantly correlated with CrCl_CG_ (r = −0.67). Lower SOFA and APACHE II scores were significantly correlated with 100% *f*T_>MIC_. Lower total bilirubin was significantly correlated with 100% *f*T_>4xMIC_. None of these covariates correlated with target attainment, or showed significant correlation with CrCl_CG_ and age, except for SOFA score, which was correlated with CrCl_CG_ (r = −0.45).

Table 3 illustrates PK/PD target attainment according to CEFu predicted by each of the five protein binding models investigated in this study. All models, except the model developed in the present study and the Bos model, showed significant disagreement in predicted vs. measured target attainment for the 100% fT_>4xMIC_ target.

## 3. Discussion

In this study, we showed suboptimal CEFu PK/PD target attainment for standard ceftriaxone 2 g q24h intermittent dosing in critically ill patients with severe CAP. Increased renal function was identified as the major driver for target non-attainment. Moreover, we showed that CEFu can be predicted from CEFt with acceptable accuracy and precision when the appropriate protein binding model is selected. As a result, the prediction of CEFu, based on the appropriate model, had no major impact on the decision of PK/PD target attainment. 

In accordance with previous PK studies in septic [9], general ICU [10,11,12] or non-ICU [18] patients, PK/PD target attainment was suboptimal for ceftriaxone 2 g q24h in patients with severe CAP. Interestingly, despite the relatively high age in our study population, renal function was still well preserved. As age and renal function are closely correlated, renal function is identified as the major driver of PK/PD target attainment. A multivariate regression analysis with CEFu as a continuous outcome variable confirmed CrCl_CG_ as the only significant covariate influencing CEFu exposure in this study (results not shown). Two recently published population PK models also identified CrCl_CG_ as a major driver for ceftriaxone clearance [10,18]. In critically ill patients, CrCl_CG_ has been shown to be biased as compared to urinary CrCl, considered the reference method to monitor renal function in the ICU [22]. However, urinary CrCl was unavailable in this study, hence we used a renal estimator, which is still often used in the ICU. Although estimated glomerular filtration according to the Chronic Kidney Disease Epidemiology equation (eGFR_CKD-EPI_) has been shown to be more accurate in ICU patients than CrCl_CG_ [22], we selected CrCl_CG_ to allow direct comparison with recent studies. A sensitivity analysis including eGFR_CKD-EPI_ instead of CrCl_CG_ led to the same results and conclusions. The median (IQR) eGFR_CKD-EPI_ was 83 mL/min/1.73 m^2^ [63, 96]. Recently, an eGFR_CKD-EPI_ cut-off of 96.5 mL/min/1.73 m^2^ was suggested to identify augmented renal clearance in ICU patients [22]. Interestingly, this suggests that the upper quartile (i.e., eGFR_CKD-EPI_ > 96 mL/min/1.73 m^2^) probably showed augmented renal clearance. Increased and/or alternative dosing regimens have been proposed for ceftriaxone to optimize PK/PD target attainment [10,11]. Still, 2 g q24h remains a frequently used standard dosing regimen [10,11,23]. This may be due to a lack of clear dose optimization strategies for ceftriaxone (and β-lactams overall) in patients with augmented renal clearance. The definite involvement of renal function, and especially augmented renal clearance, in ceftriaxone PK/PD target attainment should be confirmed by population PK modelling. Dosing simulations could then be performed to develop an optimized dosing regimen for ceftriaxone in patients with severe CAP. Unbound fractions for ceftriaxone in our study population (median 17%) are comparable to values observed in similar populations [18,21,24]. On the other hand, unbound fractions are lower than those reported by Schleibinger et al. in critically ill patients (median (IQR): 33% (20.2–44.5)) [16]. This is probably due to higher median albumin values in our population (29.5 vs. 22.6 g/L). The broader range of unbound fractions demonstrates the large variability of ceftriaxone protein binding in critically ill patients, compared to the range of unbound fractions reported in healthy volunteers (4–17%) [15]. Nonlinear saturable protein binding best described the relationship between CEFt and CEFu. Albumin concentration is the only covariate that explains a significant part of the variability in protein binding in the present study. This confirms ceftriaxone protein binding relationships described in previous PK studies [10,18]. Although bilirubin has been mentioned to influence this relationship [16], we did not find any significant influence. This is probably due to a small range of low total bilirubin values (range 0.18–1.29 mg/dL) in the present study, insufficient to displace ceftriaxone from albumin binding sites [25,26].

Concerns have been raised about the accuracy of predicting CEFu [21]. We evaluated existing methods to predict CEFu, and found that for most of them, CEFu was significantly biased in our study population. In accordance with Wong et al., our results confirm significant underestimation when predicting CEFu using the 89.5% fixed average protein binding or saturable concentration-dependent protein binding data [21]. This is an important finding to consider when interpreting results from studies that used such methods, as the predicted CEFu values probably underestimate real CEFu values. On the contrary, for the Bos and Leegwater models, there seems to be a trend towards negative bias, implying an overestimation of CEFu. It is noteworthy that these models were developed in different populations. This probably explains the significant bias and disagreement in our population. The model developed in this study predicts CEFu with acceptable bias and precision and has no major impact on clinical decisions (i.e., PK/PD target attainment). This is not unexpected as the model was developed based on the same data. This illustrates that CEFu might be accurately predicted from total concentrations, although the protein binding model used should be carefully selected as it should probably be developed in a similar population. This needs to be externally validated in future studies, which ideally measure CEFu over a broader range of albumin and total bilirubin values. The prediction of CEFu from CEFt substantially decreases the turn-around time and cost, as CEFu does not need to be measured. Hence, this could allow for the more efficient implementation of ceftriaxone TDM in clinical practice.

This study has several limitations that need to be acknowledged. First, the limited sample size precluded any robust multivariable analysis to identify independent predictors for target attainment. Nevertheless, this observational study provides a clear picture of suboptimal ceftriaxone PK/PD target attainment in patients with severe CAP. In accordance with general or septic ICU patients, renal function seems to be the major driver for ceftriaxone clearance. Although not being significant for overall target attainment, total bilirubin significantly affected the attainment of 100% fT_>4xMIC_. Hence, it might be that the sample size was too small or the range too narrow to detect a consistent effect from total bilirubin. Second, there might be covariates potentially influencing protein binding that were not measured or included in this study. Immunoglobulin G (IgG) has been reported to affect unbound ceftriaxone plasma concentrations when reaching very high concentrations, or in patients with severe hypoalbuminemia [27,28]. In the present study, most patients had moderate hypoalbuminemia, and none of the patients received IgG treatment during ceftriaxone treatment. Still, we cannot exclude any potential interference from IgG in specific patients. Interestingly, only albumin, and not total protein, was found to significantly predict ceftriaxone protein binding. Additionally, our model shows similar protein binding to two recently published models [10,18]. Third, local MIC data were not available. Therefore, we choose to apply the EUCAST clinical MIC breakpoints, which also increase generalizability.

On the other hand, this study also has several strengths. First, ceftriaxone unbound fractions were measured using equilibrium dialysis, considered the gold standard [19,20]. In most published ceftriaxone PK studies, CEFu is measured using ultrafiltration, which is less time and resource consuming. It is reassuring that the results presented in this study are in line with previous findings on ceftriaxone protein binding and PK/PD target attainment based on ultrafiltration. Still, until a direct comparison has been made between ultrafiltration vs. equilibrium dialysis, one needs to be careful in interpreting and comparing CEFu obtained by ultrafiltration. Indeed, in the case of ultrafiltration, the absence of an equilibrium between the bound and unbound drug fractions may contribute to larger variability in measured concentrations. Second, this study covered a broad range of both CEFt and CEFu, due to sampling around the peak and trough concentration. This increases the generalizability of our results.

In conclusion, we showed suboptimal unbound ceftriaxone PK/PD target attainment when using a standard 2 g q24h intermittent dosing regimen in critically ill patients with severe CAP. In accordance with studies performed in general or septic ICU patients, renal function was the major driver for failure to attain predefined PK/PD targets. Besides, CEFu was reliably predicted from CEFt without major impact on clinical decisions regarding PK/PD target attainment. This suggests that, when carefully selecting a protein binding model, CEFu does not need to be measured. As a result, as only CEFt will be measured (and CEFu will be predicted), the turn-around time and cost will substantially decrease, allowing for the efficient and broad implementation of ceftriaxone dose optimization strategies. This finding needs to be confirmed, ideally over a broader range of unbound fractions.

## 4. Materials and Methods

### 4.1. Setting, Study Design and Population

We performed a prospective single-center observational cohort study on the ICUs of a tertiary-care academic hospital (UZ Leuven, Leuven, Belgium) between January 2014 and March 2018. This study was approved by the Ethics Committee Research UZ/KU Leuven (S54509). Written informed consent was obtained from the patient or the closest relative. All adult patients admitted to the ICU with pneumonia and treated with ceftriaxone were screened for eligibility. Pregnant women and patients treated with renal replacement therapy were excluded. 

### 4.2. Study Protocol

Ceftriaxone 2 g was given once daily (i.e., q24h) as an infusion over 30 min. Sampling was performed 30 min after the end of the infusion (peak) and within 60 min before the next infusion (trough). Depending on practical feasibility, sampling was performed over one or two dosing intervals during ceftriaxone therapy (early day (day 2 +/− 1) and/or late day (day 5 +/− 1)). Blood samples were collected in lithium heparinized tubes (5 mL), and immediately refrigerated (4 °C). Samples were centrifuged within max. 24 h [29], and plasma was stored at −20 °C immediately after centrifugation [30]. All plasma samples were subsequently stored at −80 °C within 24 h after sampling [31]. Samples were transferred on dry ice from the clinical to the analytical site, where they were stored at −80 °C until analysis. Total and unbound ceftriaxone concentrations were determined in these samples.

### 4.3. Bioanalysis Method

Ceftriaxone plasma concentrations were quantified using a validated ultra-performance liquid chromatography method coupled with tandem mass spectrometry. Detailed methodology is provided in Appendix A.

#### Unbound Fraction

Plasma samples were first subjected to in vitro equilibrium dialysis. The equilibrium dialysis was carried out on a HTD96b (HTDialysis, Gales Ferry, CT, USA) device using dialysis membranes with a molecular weight cut-off 12–14 kDa. The dialysis experiments were conducted at 37 °C against phosphate buffered saline (PBS) for 4 h (experimentally determined time until equilibrium) without agitation of the plate. Spiked control samples were included for 2, 4, 40 and 320 µg/mL. Upon completion of the equilibrium dialysis, solutions from both compartments (plasma and PBS) were aliquoted. Equal amounts of blank human plasma and blank PBS were added to PBS and plasma aliquots, respectively, in order to process the samples. The processed samples were subsequently analyzed as described in Appendix A.

The unbound fraction was calculated according to the following equation [32]:(3)$$\mathrm{UF}\%=\frac{{\left[\mathrm{ceftriaxone}\right]}_{\mathrm{PBS}}}{{\left[\mathrm{ceftriaxone}\right]}_{\mathrm{plasma}}}\times 100$$
where UF% stands for the percentage of unbound ceftriaxone, [ceftriaxone]_PBS_ and [ceftriaxone]_plasma_ represent the concentration of ceftriaxone at equilibrium, in the buffer compartment and plasma compartment, respectively.

### 4.4. Protein Binding Model

Mixed-effects modeling was applied to assess the relationship that best described protein binding (linear vs. nonlinear albumin binding and association with covariates). Total and unbound ceftriaxone and serum albumin concentrations were converted to units of mmol/L; molecular weight (MW) of ceftriaxone is 661.6 g/mol [33], MW of albumin is 66,500 g/mol [34]. Serum albumin was the only plasma protein taken into account as ceftriaxone is known to bind predominantly to albumin [27]. Protein binding PK parameters were, in case of linear protein binding, the linear protein binding constant and in case of nonlinear protein binding, the maximum binding capacity (Bmax), the Hill coefficient (h) and the dissociation constant (B50) (Appendix A) [35]. Between-patient variability (BPV) was estimated for the protein binding kinetics parameters. Next, the following covariates were entered, in a stepwise forward selection, into the model in an attempt to explain BPV: albumin, age, body weight, APACHE II score, SOFA score on the day of sampling, total bilirubin, serum creatinine, creatinine clearance according to the Cockcroft–Gault formula (CrCl_CG_). The choice for CrCl_CG_ as renal estimator was made to allow direct comparison with recent studies, which mostly considered CrCl_CG_ as a covariate for ceftriaxone clearance. Afterwards, stepwise backward selection was performed to check if the included covariate significantly contributed to the final model. Model selection was based on the log-likelihood (i.e., −2LL). For the forward selection, a significance level of 5% was used. For the backward selection, a stricter significance level of 1% was applied. Finally, a visual predictive check (VPC) with 1000 simulations was performed to assess the model’s capacity to predict the range of measured CEFu.

### 4.5. Agreement between Measured CEFu and Predicted CEFu

We assessed the agreement between measured CEFu and predicted CEFu using published protein binding models, and also the model built in the present study. First, CEFu was predicted using a fixed average percentage protein binding of 89.5%, corresponding to an unbound fraction of 10.5% [15]. Second, accounting for saturable concentration-dependent protein binding of ceftriaxone, CEFu was predicted using the following published model [9,36,37]:(4)$$\mathrm{CEFu}=\frac{1}{2}\;\left(-\left(\mathrm{nP}+\frac{1}{k_{\mathrm{aff}}}-\mathrm{CEFt}\right)+\sqrt{{\left(\mathrm{nP}+\frac{1}{k_{\mathrm{aff}}}-\mathrm{CEFt}\right)}^{2}+\frac{4\mathrm{CEFt}}{k_{\mathrm{aff}}}}\right)$$
where CEFu stands for unbound ceftriaxone concentration (mmol/L), nP stands for capacity constant (517 µmol/L for ceftriaxone), k_aff_ stands for binding affinity constant (0.0367 L/µmol for ceftriaxone), and CEFt stands for total ceftriaxone concentration (mmol/L).

Third, the model for CEFt built in the present study was used to predict CEFu. Finally, two published models (i.e., Bos model [18] and Leegwater model [10]), also describing nonlinear albumin dependent protein binding, were used to predict CEFu.

Agreement between the measured CEFu and CEFu predicted with each of these five methods was assessed by calculating bias (CEFu measured—CEFu predicted), relative bias (percent of mean CEFu measured) and relative root mean squared error (RMSE, percent of mean CEFu measured). Additionally, agreement was graphically assessed in a Bland–Altman plot [38]. The mean relative bias, and upper and lower limits of agreement are shown.

### 4.6. PK/PD Target Attainment

For β-lactams, unbound concentrations above the MIC or 4-fold the MIC throughout the whole dosing interval (i.e., 100% *f*T_>MIC_ and 100% *f*T_>4xMIC_) have been recommended as PK/PD target for optimal clinical outcome in ICU patients [6,11,39]. Both targets were assessed in this study. As the causative pathogen and its MIC are usually unknown upon (empirical) initiation of antimicrobial therapy, we assessed PK/PD target attainment according to a worst-case scenario. Therefore, considering the typical target pathogens and their clinical MIC breakpoint for susceptibility to ceftriaxone, a target MIC of 1 mg/L was chosen (i.e., breakpoint of Enterobacterales and *Moraxella cattarhalis* recommended by the European Committee on Antimicrobial Susceptibility Testing; http://www.eucast.org/clinical_breakpoints (accessed on 26 November 2020)) [1,2]. Hence, successful PK/PD target attainment is defined as CEFu exceeding 1 mg/L or 4 mg/L throughout the whole dosing interval, for 100% fT_>MIC_ or 100% fT_>4xMIC_, respectively. Successful target attainment in ≥90% of all dosing intervals was considered as an acceptable overall target attainment.

The correlation of patient characteristics with PK/PD target attainment was evaluated in univariate analysis. Additionally, we assessed (dis)agreement between PK/PD target attainment based on CEFu predicted using the five aforementioned models and target attainment based on CEFu measured in this study.

### 4.7. Statistical Analysis

The results are expressed as mean ± standard deviation or median (interquartile range), as appropriate. Student’s *t*-tests or Wilcoxon rank sum tests are applied, as appropriate, to investigate a potential association between patient covariates and PK/PD target attainment. All analyses of CEFt and CEFu were performed in R (version 3.5.1 or higher, R Core Team, Vienna, Austria) in the RStudio integrated development environment (version 1.3; RStudio, Inc., Boston, MA, USA). The following R packages were used: dplyr, nlme, and ggplot2. The null hypothesis of *μ_A_* = *μ_B_* was tested against a two-sided alternative hypothesis at the 5% significance level.

## Figures and Tables

**Figure 1 antibiotics-10-00557-f001:**
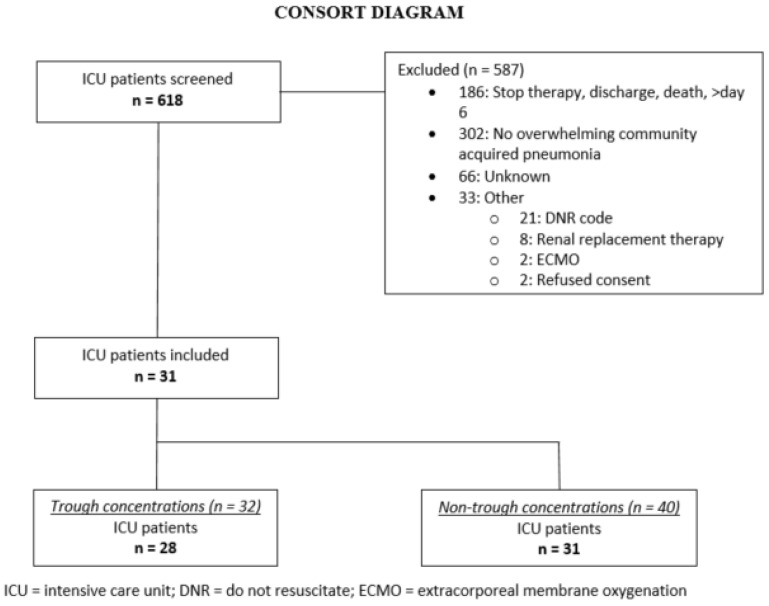
Study flow diagram.

**Figure 2 antibiotics-10-00557-f002:**
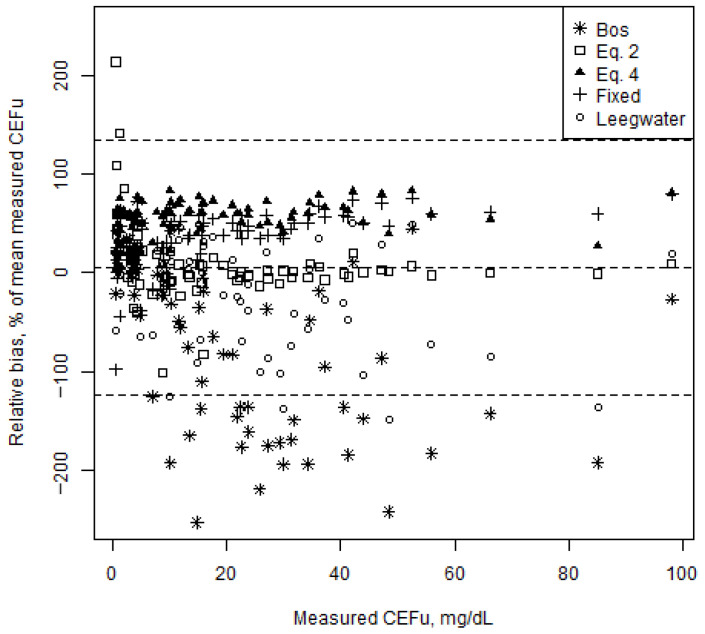
Bland–Altman plot representing the relative bias of the unbound ceftriaxone concentrations (CEFu) predicted from the five protein binding models vs. measured CEFu for all CEFu (*n* = 72). Relative biases are shown as percentage of the mean measured CEFu (*y*-axis) in function of the measured CEFu. The dashed lines represent the mean relative bias and the upper and lower limits of agreement (95% confidence intervals).

**Table 1 antibiotics-10-00557-t001:** Patient characteristics.

Parameter	Overall	On *early* Sampling Day	On *late* Sampling Day
Number of patients, *n* (%) ^a^	31 (100)	25 (81)	11 (35)
Demographics			
Sex, male, *n* (%)	19 (61)		
Age, median (IQR), years	72 (55–81)		
Body weight, median (IQR), kg	64 (60–77)		
Clinical scores			
Sequential Organ Failure Assessment score, median (IQR), *n*	6 (4–9)	8 (6–10), 15	5 (3–6), 11
Acute physiology and chronic health evaluation II score, median (IQR)	18 (16–26)		
Biochemical parameters			
Serum creatinine, median (IQR), mg/dL, *n*	0.86 (0.71–1.04)	0.91 (0.71–1.15), 25	0.8 (0.71–0.87), 11
Cockcroft–Gault equation, median (IQR), mL/min, *n*	73 (54–102)	71 (49–90), 25	87 (58–109), 11
Serum albumin, median (IQR), g/L, *n*	29.5 (26.7–31.7)	29.5 (26.8–31.6), 25	29.6 (26.2–31.5), 11
Total bilirubin, median (IQR), mg/dL, *n*	0.8 (0.25–0.6)	0.38 (0.26–0.57), 25	0.28 (0.24–0.68), 11
Sampling			
Unbound ceftriaxone pre-dose concentration ≥1 mg/L, *n* (%)	26/32 (81)	17/21 (81)	9/11 (81.8)
Unbound ceftriaxone pre-dose concentration ≥4 mg/L, *n* (%)	15/32 (47)	10/21 (47.6)	5/11 (45.5)

^a^ Five patients were sampled on both days. IQR: interquartile range, *n*: number of patients.

**Table 2 antibiotics-10-00557-t002:** Agreement between measured and predicted unbound concentration of ceftriaxone.

	CEFu Predicted mg/L, Median (IQR)	fu % Predicted, Median (IQR)	Bias mg/L, Median (IQR)	Relative Bias, % of CEFu, Median (IQR)	Relative RMSE, % of CEFu	*p*-Value ^b^
Fixed average protein binding	8.3 (3.4–13.9)	10.5 ^a^	5.6 (0.6–12)	38.2 (19.2–50.6)	86.1	<0.0001
Predicted saturable concentration-dependent protein binding, Equation (4)	5 (1.8–10)	6.3 (5.5–7.6)	8.4 (2.2–15.7)	59.2 (50.9–68)	89	<0.0001
Predicted protein binding, present study, Equation (2)	15.5 (4.2–30.4)	17.9 (12.3–22.6)	0.05 (−0.8–1.1)	0.4 (−5.8–12.6)	14.2	0.627
Predicted protein binding, Bos	21.8 (3.5–62.1)	28.2 (11.7–45.3)	−6.2 (−35.9–0.1)	−35.9 (−143.7–4.8)	195.3	<0.0001
Predicted protein binding, Leegwater	11.5 (3.2–33.4)	14.7 (10–25.2)	0.3 (−10.1–1.5)	10.2 (−49.5–29.3)	109.2	0.350

CEFu: unbound ceftriaxone concentration; fu: unbound ceftriaxone fraction; 100% *f*T_>MIC_: free concentration above the minimum inhibitory concentration over the whole dosing interval (100%); 100% *f*T_4x>MIC_: free concentration above 4 times the minimum inhibitory concentration over the whole dosing interval (100%); IQR: interquartile range. ^a^ no median (IQR) due to a fixed protein binding coefficient; ^b^ null hypothesis = difference between CEFu measured and CEFu predicted is equal to 0.

**Table 3 antibiotics-10-00557-t003:** Agreement for PK/PD target attainment based on measured and predicted unbound concentration of ceftriaxone.

	100% fT_>MIC_, *n* (%)	*p*-Value ^a^	100% fT_>4xMIC_, *n* (%)	*p*-Value ^a^
Fixed average protein binding	26 (81.2)	1.000	11 (34.4)	0.044
Predicted saturable concentration-dependent protein binding, Equation (4)	23 (71.2)	0.083	5 (15.6)	0.0007
Predicted protein binding, present study, Equation (2)	24 (75)	0.572	13 (40.6)	1.000
Predicted protein binding, Bos	25 (78.1)	0.325	13 (40.6)	0.325
Predicted protein binding, Leegwater	25 (78.1)	0.325	11 (34.4)	0.044

100% *f*T_>MIC_: free concentration above the minimum inhibitory concentration over the whole dosing interval (100%); 100% *f*T_4x>MIC_: free concentration above 4 times the minimum inhibitory concentration over the whole dosing interval (100%). ^a^ Null hypothesis = no difference (i.e., agreement) in PK/PD target attainment based on measured vs. predicted CEFu.

## Data Availability

The data presented in this study are available on request from the corresponding author. The data are not publicly available due to privacy and ethical reasons.

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
