# Peer review of "Pharmacokinetic/Pharmacodynamic Target Attainment Based on Measured versus Predicted Unbound Ceftriaxone Concentrations in Critically Ill Patients with Pneumonia: An Observational Cohort Study"

_antibiotics, 2021, doi:10.3390/antibiotics10050557_

Round 1

Reviewer 1 Report

Pharmacokinetic/pharmacodynamic target attainment based on measured versus predicted unbound ceftriaxone concentrations in critically ill patients with pneumonia: an observational cohort study

Reviewer comments

General comments:

This paper describes the impact of protein binding on ceftriaxone PK/PD target attainement in ICU population, using an interesting approach comparing observed unbound ceftriaxone concentrations vs predicted from various PK models, including one PK model developed in this study. However, some issues have to be corrected before publication.

English language have to be improve in the whole manuscript, especially in the abstract and in the introduction sections.

As described in specific comments, presentation of results have also to be improve.

Specific comments:

  • Abstracts
    • “PK/PD target attainement was defines as >90%”? 100 % is described in the method section?
    • “CEFu”: abbreviation has to be defined in the abstract.

  • Introduction 
    • Numerous studies have described ceftriaxone PK in ICU population, including data on unbound concentration. Specific contribution of present study has to be specify in this section.

  • Results
    • “non through concentration “ : correspond to peak concentrations and/or through concentrations with incorrect sample time ?
    • “ 36 dosing intervals” mean 36 ceftriaxone treatments ? This is unclear.
    • Overall, presentation of concentrations and population results is unclear :
      • notion of “through”, “non through”, “early or late inclusion” are not clear and Figure 1 could maybe be improve.
      • Patient characteristics in Table 1 are presented owing to sampling day. The goal is to compare data between sample days??
    • Tables : presentation have to be improve (references, legend, standardization between tables, ...).

  • Discussion

Why authors did not presented dose simulations in this study, using developed PK model and optimization focus on renal function? This point have to be discuss.

Reviewer 2 Report

The topic of this manuscript is interesting. The reviewer feels this manuscript can be accepted after some minor amendments.

(1) Why the sample size is so small?

(2) The gender is not balanced.

(3) The accuracy, precision, stability of the bioanalytical method is missing. 

Reviewer 3 Report

The authors should consider the followings:

  1. What are the possible binding partner of ceftriaxone (other than albumin)? The authors should mention the limitation of such bindings, and the references.
  2. Please state whether any incurred sample re-analyses were performed. What were the acceptance criteria that. Please explain why it was not performed.
  3. What are the long term stability of the plasma samples?
  4. Did the author perform any stability studies of the equilibrium dialysis?
  5. In section 4.3.1, did the author perform any studies to verify the use of 4 hour as equilibrium dialysis time; otherwise, why choose this timepoint?
  6. In section 4.2, please state the stability data of ceftriaxone in whole blood.
  7. If a second major protein was bound to ceftriaxone in the author’s model, would the ultimate results still fall within the same test hypothesis?
  8. In the abstract, please state the full name of your abbreviation, i.e. CEFu.
  9. In Figure 1, please explain why “ICU patients included n =31”, while that the lower boxes showed n=32, and n=40? Please present the data in clearly.
  10. In section 4.2, please specific how long the samples stayed in whole blood, before centrifugation. Would there be any deviation between the early centrifuge vs the late centrifuge samples, since up to 24 is considered a long duration?
  11. As referred to section 4.2, please specify whether any documentation record available, for the sample transfer between sites (clinical to analytical site?) Were the sample maintained below -70’c during transfer?
  12. In section 4.3.1, please mention whether any agitation (plate shaking) required during equilibrium dialysis, of the plate.
  13. In section S1, please mention the grading of standard(s) used in the study, of pharmaceutical secondary (or primary) standards?
  14. Were the analytical lab performing the said bioanalytical assay, accredited? And was it within their accredited scope?
  15. Please state clearly in the abstract and conclusion for the novelty of this study.
